# A Multiplex PCR-Based Next Generation Sequencing-Panel to Identify Mutations for Targeted Therapy in Breast Cancer Circulating Tumor Cells

**André Franken, Mahdi Rivandi, Liwen Yang, Bernadette Jäger, Natalia Krawczyk, Ellen Honisch, Dieter Niederacher, Tanja Fehm † and Hans Neubauer \*,†**

Department of Obstetrics and Gynecology, University Hospital and Medical Faculty of the Heinrich-Heine University Duesseldorf, 40225 Duesseldorf, Germany; andre.franken@med.uni-duesseldorf.de (A.F.); mahdi.rivandi@med.uni-duesseldorf.de (M.R.); liwen.yang@med.uni-duesseldorf.de (L.Y.); bernadette.jaeger@med.uni-duesseldorf.de (B.J.); natalia.krawczyk@med.uni-duesseldorf.de (N.K.); Ellen.Honisch@med.uni-duesseldorf.de (E.H.); niederac@med.uni-duesseldorf.de (D.N.); tanja.fehm@med.uni-duesseldorf.de (T.F.)

\* Correspondence: hans.neubauer@med.uni-duesseldorf.de

† Both authors share last authorship.

**Abstract:** Targeted therapy has become the preferred approach to treat most cancers, including metastatic breast cancer. Using liquid biopsies, which can act as a dynamic diagnostic tool, is an appealing concept to identify effective therapies. In order to identify mutations from circulating tumor cells (CTCs) on single cell level, we have developed a multiplex PCR-based next generation sequencing-panel. The CTCs were enriched using the CellSearch system and isolated by micromanipulation followed by whole genome amplification of their DNA. Afterwards, mutation hotspot regions in the *PIK3CA*, the *ESR1*, the *AKT1*, and the *ERBB2* genes were amplified and barcoded. Sequencing was performed on a MiSeq system. The assay was validated with cells from various cell lines displaying the expected mutations. Mutations that provide the basis for potential targeted therapies were detected in 10 out of 13 patients in all analyzed genes. In four patients, mutations in more than one gene were observed—either in the same cell or in different cells, suggesting the presence of different tumor cell clones, which might be targeted with combination therapies. This assay is a time and cost effective tool to investigate the most relevant genomic positions indicative for targeted therapies in metastatic breast cancer. It can support therapy decision to improve the treatment of cancer patients.

**Keywords:** breast cancer; circulating tumor cells; liquid biopsy; metastatic breast cancer; panel sequencing; targeted therapy; single cell analysis

## 1. Introduction

In the last decades, targeted therapy has become the preferred approach to treat most cancers. However, obtaining information to choose a targeted therapy is challenging: biopsies of recurrent or metastatic lesions are invasive and cannot be performed when clinical conditions have worsened or when a tumor is inaccessible [1]. Furthermore, the genomic profile of biopsy tissues provides a picture limited to a single point in space and time, and may, thus, under-represent intratumoral heterogeneity [2]. The predictive utility of tissue biopsies is limited by such factors, worsened still by the continuous evolution of the tumor in response to endogenous and exogenous selective pressures [3]. Deriving information about the primary tumor or metastatic lesions from liquid biopsies, which may act as a dynamic diagnostic tool, is an appealing concept to overcome this challenge [4,5].

Potential biomarkers are circulating tumor cells (CTCs) and circulating tumor DNA (ctDNA). CTCs are shed into the blood by tumor tissue and are commonly considered as precursor cells for metastasis formation [6,7]. ctDNA as part of cell free DNA describes tumorous DNA circulating in the blood, which has been released from degraded tumor cells [8].

In breast cancer, clinical utility has been demonstrated for the therascreen *PIK3CA* RGQ PCR kit, which detects *PIK3CA* mutations in a tissue biopsy and/or inctDNA obtained from a liquid biopsy in patients with an advanced or metastatic tumor [9]. The *PIK3CA* gene encodes for the p110α subunit of the phosphoinositide-3-kinase (PI3K) which promotes cell proliferation and survival by activation of the PI3K/AKT signaling pathway [10]. Mutations, mostly located in its catalytic domain, can lead to an activation of downstream pathways [11]

Further mutations that could act as targets for therapy in breast cancer are located in the *ESR1*, the *AKT1* and the *ERBB2* genes: the *ESR1* gene codes for the estrogen receptor α (ERα), a nuclear hormone receptor that is involved in the regulation of gene expression affecting cellular proliferation and differentiation [12]. Mutations in the ligand binding domain of ERα lead to a conformational change of the protein and to a ligand-independent ERα activation, mostly acquired in resistance to estrogen deprivation therapy such as aromatase inhibition [13,14]. AKT1 is a member of the serine-threonine kinase class and plays a key role in cellular processes, including growth, proliferation, survival, and angiogenesis. Certain *AKT1* mutations lead to hyperactivation of the mTOR pathway [15]. The receptor tyrosine-protein kinase ERBB2 (HER2/neu (HER2)) activates the RAS/MAPK, PI3K/AKT, and JAK/STAT signaling pathways to promote cell proliferation and survival. Mutations in the hotspot regions coding for the extracellular domain and the kinase domain lead to an activation of the protein and its downstream pathways [16].

In order to identify mutations for targeted therapy in breast cancer using CTCs, which have been enriched using the CellSearch system, we developed a multiplex PCR-based next generation sequencing (NGS)-panel. The CellSearch system has been approved by the Food and Drug Administration (FDA) for detection of CTCs in patients with metastatic breast, prostate, and colorectal cancer.

## 2. Materials and Methods

### 2.1. Patients

We analyzed CTCs from 13 metastatic breast cancer patients. Patients were selected from the Augusta study cohort collected at the Department of Obstetrics and Gynecology, Duesseldorf, Germany (approved by the Ethics Committee of the Medical Faculty of the Heinrich Heine University Düsseldorf; Ref-No: 3430). All patients gave their informed consent for the use of their blood samples for CTC analysis and translational research. Patients' characteristics were anonymized by using sample identifiers. Clinical patient data are shown in Table 1.

### 2.2. Enrichment and Enumeration of CTCs

Enrichment and enumeration of CTCs was performed using the CellSearch Circulating Epithelial Cell Kit (Menarini, Florence, Italy) according to manufacturer's instructions. Briefly, blood was collected into CellSave Preservative Tubes (Menarini) and was processed within 96 h. 7.5 mL blood was used for enumeration of CTCs. Enrichment is based on immunomagnetic ferrofluid conjugated with an epithelial cell adhesion molecule-directed antibody. Subsequent characterization of CTCs was performed using immunofluorescent staining directed against cytokeratins to identify CTCs, CD45 to exclude leucocytes and DAPI to confirm nucleo-morphological integrity.

### 2.3. CTC Isolation

Single CTCs were isolated from the CellSearch cartridge by micromanipulation with the CellCelector (ALS, Jena, Germany). Cells were deposited in PCR tubes to perform whole genome amplification (WGA) as described previously [17].

## 2.4. Whole-Genome Amplification

DNA of single isolated cells was amplified by WGA with the Ampli1 WGA Kit (Menarini) according to the manufacturer's protocol. Afterwards DNA integrity was determined with the Ampli1 QC Kit (Menarini).

**Table 1.** Patient characteristics.

| Characteristics | Total | in % |
|---|---|---|
| Patients | 13 | 100 |
| Age at blood draw | | |
| Mean | 61.2 | |
| Median | 61 | |
| Range | 46–78 | |
| Tumor size | | |
| pT1 | 4 | 30.8 |
| pT2–4 | 8 | 61.5 |
| na | 1 | 7.7 |
| Nodal status at time of diagnosis | | |
| 0 | 4 | 30.8 |
| 1–3 | 7 | 53.8 |
| na | 2 | 15.4 |
| Metastasis status at time of diagnosis | | |
| 0 | 10 | 77.0 |
| 1 | 2 | 15.4 |
| na | 1 | 7.7 |
| Histology | | |
| NST | 8 | 61.5 |
| Invasive-lobular | 5 | 38.5 |
| Grading | | |
| 1–2 | 9 | 69.2 |
| 3 | 2 | 15.4 |
| na | 2 | 23.1 |
| Subtype | | |
| Luminal | 11 | 84.6 |
| Triple negative | 2 | 15.4 |
| CTC count (per 7.5 mL blood) | | |
| Median | 94 | |
| Range | 6–15,340 | |

NST, invasive carcinoma of no special type; na, data not available; CTC, circulating tumor cell.

## 2.5. Multiplex PCR-Based Next Generation Sequencing

DNA fragments were amplified and barcoded in two steps using KAPA2G Fast Multiplex Mix (Merck, Darmstadt, Germany). In the first step, fragments coding for mutation hotspot regions were amplified with gene specific primers (Table 2) containing primer binding sites for universal Multiplicom MID Dx primers (Agilent technologies, Santa Clara, CA, USA) in a multiplex PCR. 1 μL WGA product and in total 0.5 μM gene specific primers were applied (concentrations of all primers are shown in Table S1). The concentration of each primer is shown in Table S1. In the second step, Multiplicom MID Dx primers were used to attach unique barcodes for sequencing to the PCR products. 0.5 μL of Multiplicom MID Dx primers and 0.5 μL of first round PCR product were used. The applied PCR protocols are listed in Table S2.

Final PCR products were purified using Agencourt AMPure XP beads (Beckmann Coulter, Brea, CA, USA) and quantified with the Qubit 2.0 Fluorometer (Thermo Fisher Scientific, Waltham, MA, USA) using the Broad Range Quant-iT dsDNA Assay Kit (Thermo Fisher Scientific).

DNA libraries of 0.0143 pM per sample were sequenced on a MiSeq system (Illumina, San Diego, CA, USA) with 151 bp paired end reads.

FASTQ files were analyzed with the Galaxy web platform, using the public server at usegalaxy.org (last access for the Galaxy web platform: March 2020) [18]. Reads were aligned to the human reference genome (hg38) with BWA-MEM (Galaxy Version 0.7.17.1) [19]. Standard parameters were applied. Aligned sequences were screened for mutations using the IG Viewer (Version 2.3.25) [20]. Identified mutations were covered by >100× and had a variant allele frequency (VAF) >12.5%.

As controls, DNA from the reference cell lines mentioned below as well as a negative control was included in every run.

**Table 2.** Primer overview.

| | Forward Primer | Reverse Primer |
|---|---|---|
| *PIK3CA* Exon 5 | 5′ AAGACTCGGCAGCATCTCCAGC ATTTCCACAGCTACACCA 3′ | 5′ GCGATCGTCACTGTTCTCCAGAT GTTCTCCTAACCATCTGA 3′ |
| *PIK3CA* Exon 10 | 5′ AAGACTCGGCAGCATCTCCAGGG AAAATGACAAAGAACAG 3′ | 5′ GCGATCGTCACTGTTCTCCAATTT TAGCACTTACCTGTGAC 3′ |
| *PIK3CA* Exon 21 | 5′ AAGACTCGGCAGCATCTCCATTGA TGACATTGCATACATTCG 3′ | 5′ GCGATCGTCACTGTTCTCCAGTG GAAGATCCAATCCATTT 3′ |
| *ESR1* Exon 5 | 5′ AAGACTCGGCAGCATCTCCAT TGACCCTCCATGATCAGGT 3′ | 5′ GCGATCGTCACTGTTCTCCAGC TACTCCTAAGCTACAGCC 3′ |
| *ESR1* Exon 7 | 5′ AAGACTCGGCAGCATCTCCATCT CTCACTCTCTCTCTGCG 3′ | 5′ GCGATCGTCACTGTTCTCCAGATGT GGGAGAGGATGAGGA 3′ |
| *ESR1* Exon 8 | 5′ AAGACTCGGCAGCATCTCCAAGT AGTCCTTTCTGTGTCTTC 3′ | 5′ GCGATCGTCACTGTTCTCCAAAT GCGATGAAGTAGAGCCC 3′ |
| *AKT1* Exon 3 | 5′ AAGACTCGGCAGCATCTCCAGTA GAGTGTGCGTGGCTC 3′ | 5′ GCGATCGTCACTGTTCTCCACCC CAAATCTGAATCCCGAG 3′ |
| *ERBB2* Exon 8 | 5′ AAGACTCGGCAGCATCTCCAGGC TACATGTTCCTGATCTCC 3′ | 5′ GCGATCGTCACTGTTCTCCAG GGTCTGAGGAAGGATAGGA 3′ |
| *ERBB2* Exon 18 | 5′ AAGACTCGGCAGCATCTCCAA AGTACACGATGCGGAGACT 3′ | 5′ GCGATCGTCACTGTTCTCCAACCTT CACCTTCCTCAGCTC 3′ |
| *ERBB2* Exon 19 | 5′ AAGACTCGGCAGCATCTC CAATCCTCCTCTTTCTGCCCAG 3′ | 5′ GCGATCGTCACTGTTCTCCAAGTCTA GGTTTGCGGGAGTC 3′ |
| *ERBB2* Exon 20 | 5′ AAGACTCGGCAGCATCTC CATGGTTTGTGATGGTTGGGAG 3′ | 5′ GCGATCGTCACTGTTCTCCAGA CATGGTCTAAGAGGCAGC 3′ |

*2.6. Validation Experiments with Spiked Cells*

For validation of the workflow, 200 cells of the cell lines SK-BR-3, T47-D, and MCF7 (ATCC, Manassas, United States; catalog numbers: SK-BR-3: HTB-30, T-47D: HTB133, and MCF7: HTB-22) as well as long term cultured CTCs were spiked into 7.5 mL healthy donor blood. For detachment from the culture flask, an enzyme-free cell dissociation buffer (Thermo Fisher Scientific) was used. The tumor cells were enriched using the CellSearch, isolated using the CellCelector and the genome was amplified with the Ampli1 WGA Kit similarly to CTCs.

The SK-BR-3, T47-D, and MCF7 cells were cultured in RPMI 1640 containing 10% fetal calf serum, 25 mmol/L HEPES and 1% Penicillin-Streptomycin (all Thermo Fisher Scientific). Cells were grown at 37 °C in a humidified atmosphere with 5% $CO_2$ and were authenticated via short tandem repeat analysis. CTCs were cultured as previously described in low attachment plates (Corning, New York, NY, USA) in RPMI 1640 medium supplemented with 1× B27 (Thermo Fisher Scientific), 20 ng/mL hEGF (Merck), 20 ng/mL FGF (Merck), and 1% Penicillin-Streptomycin in a humidified atmosphere with 5% $CO_2$ and 4% $O_2$ [21]. Cultured cells were regularly tested negative for Mycoplasma infection.

*2.7. Statistical Analysis*

Statistical analyses were performed using GraphPad Prism (Graphpad Software, San Diego, CA, USA). *p*-values < 0.05 were considered statistically significant.

## 3. Results

### 3.1. Development of the Assay

We aimed to develop a multiplex PCR-based NGS panel to identify mutations in CTCs relevant for targeted therapies in metastatic breast cancer, considering that the genomic DNA of single CTCs was amplified using an MseI restriction-based method. We included hotspot regions in the *PIK3CA*, *ESR1*, *AKT1*, and *ERBB2* genes frequently mutated in breast cancers. In the *PIK3CA* gene, hotspot regions within the exons 5, 10, and 21 were covered. In the *ESR1* gene, the hotspot regions in exons 5, 7, and 8 were analyzed. In the *AKT1* gene, we investigated the hotspot regions in exon 3 and in the *ERBB2* gene our assay covers the hotspot regions in exons 8, 18, 19, and 20. For each amplicon, a panel of primers was tested and the most specific primer combination was chosen for the multiplex PCR (Table 2, Figure S1). Primer concentrations in the first PCR were titrated resulting in an equal ratio for each fragment of 7.0% ± 1.4% to 10.2% ± 0.8% after sequencing (Figure S2). In total, we achieved an on target rate of 95.3% ± 0.9% of all reads.

Next, a group of CTCs of different quality determined by Ampli1 quality control was sequenced and the obtained coverage was compared. For cells with highest amplified DNA quality showing 4 bands in the quality control PCR a mean coverage of 10.2 ± 1.3 fragments per cell was observed (Figure 1). The coverage correlated with the quality determined with the quality control PCR. Cells displaying 3 or 4 bands in the quality control PCR had a significantly increased coverage compared to those with less than three bands ($p < 0.0001$ determined by two-tailed *t*-test). Thus, we focused on cells displaying at least 3 bands in the quality control PCR for further analysis to yield reliable results.

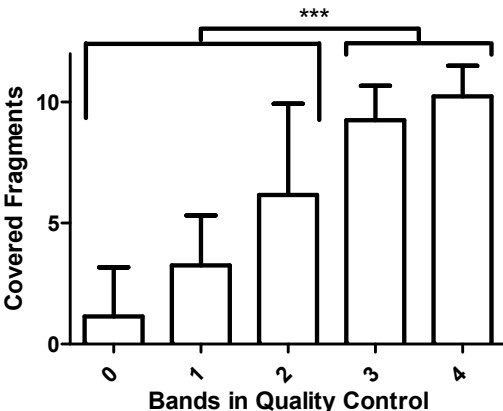

**Figure 1.** The number of covered fragments correlates with quality of WGA products. The *p*-value ($p < 0.0001$) was determined by two-tailed *t*-test. Error bars show standard deviation. For each group 10 to 30 cells were analyzed.

### 3.2. Validation of the Assay

To validate the assay, we performed spike in experiments using cells from four cell lines exhibiting different mutations. Those cells were processed applying the same workflow used later for clinical samples. As a negative control, cells from the Her2-enriched breast cancer cell line Sk-Br-3 harboring no mutations in the analyzed regions were analyzed. Cells from the luminal breast cancer cell lines T47-D and MCF7 were expected to show the *PIK3CA* mutations H1047R and E545K, respectively. Further, long term cultured CTCs derived from the DLA product of a metastatic luminal breast cancer patient harboring the *ESR1* mutation E380Q and the *AKT1* mutation E17K were analyzed. All investigated cells displayed the expected mutations demonstrating the reliability of our assay (Figure 2).

| Cell line | Cell ID | PIK3CA | | ESR1 | AKT1 |
|---|---|---|---|---|---|
| | | E545K | H1047R | E380Q | E17K |
| Sk-Br-3 | 1 | | | | |
| | 2 | | | | |
| | 3 | | | | |
| | 4 | | | | |
| | 5 | | | | |
| | 6 | | | | |
| T47-D | 1 | | 21% | | |
| | 2 | | 24% | | |
| | 3 | | 100% | | |
| MCF-7 | 1 | 49% | | | |
| | 2 | 58% | | | |
| | 3 | 75% | | | |
| | 4 | 57% | | | |
| | 5 | 41% | | | |
| CTC (LTC) | 1 | | | 100% | 50% |
| | 2 | | | 100% | 100% |
| | 3 | | | 100% | 41% |

**Figure 2.** Sequencing data of cells from reference cell lines. Displayed values show the variant allele frequency. Empty boxes indicate that no mutation was detected. LTC, long term cultured.

### 3.3. Analysis of CTCs from Metastatic Breast Cancer Patients to Identify Targeted Therapies

After validation with cell line cells we analyzed CTCs from 13 metastatic breast cancer patients. 11 of those patients had a primary tumor of luminal subtype and two patients were diagnosed with triple negative breast cancer. CellSearch analyses determined 6 to 15,340 CTCs per 7.5 mL blood. The median CTC count was 94 per 7.5 mL blood. For our study, 1 to 15 CTCs per patient were analyzed.

In seven patients, mutations in the *PIK3CA* gene were identified. Five of those mutations are well-described mutations located in the nucleotide triplets coding for the amino acid N345, E545, and H1047, respectively. *ESR1* mutations were detected in CTCs from two patients. One of those in the coding triplet for L536 is located in a well-known hotspot region. In CTCs from three patients, the *AKT1* mutation E17K was found, and in CTCs from five patients we discovered mutations in the *ERBB2* gene, including hotspot mutations in the nucleotide triplets coding for S310, L755, and V777 (Figure 3).

Mutations were detected in 16.7% (*ERBB2* A775T, patient 4) to 100% (e.g., *PIK3CA* H1047R, patient 8) of the CTCs analyzed per patient. In two cases, mutations were detected in two genes (*AKT1* and *ERBB2*, patient 6; *PIK3CA* and *ERBB2*, patient 8) per patient, in 1 patient we found mutations in three genes (*PIK3CA*, *ESR1*, and *AKT1*, patient 3). Those mutations either occurred in different CTCs (e.g., *AKT1* and *PIK3CA* mutations in CTCs from patient 3) or together in one cell (e.g., *PIK3CA* and *ERBB2* mutations in patient 8). One CTC from patient 3 harbored two activating mutations within the same gene (*PIK3CA* N345K and H1047R). Most mutations were detected with frequencies of 100% (e.g., *AKT1* E17K, patient 2) or about 50% (e.g., *PIK3CA* H1047R, patient 8).

| Patient ID | CTC ID | PIK3CA | | | | | | ESR1 | | AKT1 | ERBB2 | | | | |
|---|---|---|---|---|---|---|---|---|---|---|---|---|---|---|---|
| | | M318T | N345K | E545K | E547K | H1047L | H1047R | L536P | L541Q | E17K | S310F | L313P | L755S | A775T | V777L |
| Patient 1 | 1 | | | | | | | | | | | | | | |
| | 2 | | | | | | | | | | | | | | |
| | 3 | | | | | | | | | | | | | | |
| | 4 | | | | | | | | | | | | | | |
| | 5 | | | | | | | | | | | | | | |
| Patient 2 | 1 | | | | | | | | | 100% | | | | | |
| | 2 | | | | | | | | | 100% | | | | | |
| | 3 | | | | | | | | | | | | | | |
| | 4 | | | | | | | | | 100% | | | | | |
| | 5 | | | | 16% | | | | | 100% | | | | | |
| | 6 | | | | | | | | | 100% | | | | | |
| | 7 | | | | | | | | | 100% | | | | | |
| | 8 | | | | | | | | | 100% | | | | | |
| | 9 | | | | | | | | | 100% | | | | | |
| Patient 3 | 1 | | | | | | | | | 100% | | | | | |
| | 2 | | | | | | 100% | 30% | | | | | | | |
| | 3 | | 100% | | | | | | | | | | | | |
| | 4 | | | | | | | | | | | | | | |
| | 5 | | 100% | | | | 100% | | | | | | | | |
| Patient 4 | 1 | | | | | | | | | | | | | | |
| | 2 | | | | | | | | | | | | | | |
| | 3 | | | | | | | | | | | | | | |
| | 4 | | | | | | | | | | | | | 34% | |
| | 5 | | | | | | | | | | | | | | |
| | 6 | | | | | | | | | | | | | | |
| Patient 5 | 2 | | | | | | 80% | | | | | | | | |
| | 3 | | | | | | | | | | | | | | |
| | 4 | | | | | | 43% | | | | | | | | |
| Patient 6 | 1 | | | | | | | | | | | | 25% | | |
| | 2 | | | | | | | | | | | | 70% | | |
| | 3 | | | | | | | | | 100% | | | 74% | | |
| | 4 | | | | | | | | | | | | 50% | | |
| | 5 | | | | | | | | | 38% | | | | | |
| | 6 | | | | | | | | | | | | | | |
| | 7 | | | | | | | | | | | | 49% | | |
| Patient 7 | 1 | | | | | | | | | | | | | | 46% |
| Patient 8 | 1 | | | | | | 51% | | | | | | | | |
| | 2 | | | | | | 51% | | | | | | | | |
| | 3 | | | | | | 52% | | | | 40% | | | | |
| | 4 | | | | | | 34% | | | | 32% | | | | |
| | 5 | | | | | | 49% | | | | 36% | | | | |
| | 6 | | | | | | 51% | | | | 35% | | | | |
| | 7 | | | | | | 50% | | | | | | | | |
| Patient 9 | 1 | | | | | 79% | | | | | | 43% | | | |
| | 2 | 13% | | | | 50% | | | | | | | | | |
| | 3 | | | | | 34% | | | | | | | | | |
| | 4 | | | | | | | | | | | | | | |
| | 5 | | | | | | | | | | | | | | |
| | 6 | | | | | 31% | | | | | | | | | |
| | 7 | | | | | 22% | | | | | | | | | |
| | 8 | | | | | 52% | | | | | | | | | |
| | 9 | | | | | 59% | | | | | | | | | |
| | 10 | | | | | 61% | | | | | | | | | |
| | 11 | | | | | 60% | | | | | | | | | |
| | 12 | | | | | 54% | | | | | | | | | |
| | 13 | | | | | 85% | | | | | | | | | |
| | 14 | | | | | | | | | | | | | | |
| | 15 | | | | | | | | | | | | | | |
| Patient 10 | 1 | | | | | | | | | | | | | | |
| | 2 | | | | | | | 100% | | | | | | | |
| | 3 | | | | | | | | | | | | | | |
| | 6 | | | | | | | | | | | | | | |
| Patient 11 | 1 | | | | | | | | | | | | | | |
| | 2 | | | | | | | | | | | | | | |
| | 3 | | | | | | | | | | | | | | |
| | 4 | | | | | | | | | | | | | | |
| | 5 | | | | | | | | | | | | | | |
| Patient 12 | 1 | | | 100% | | | | | | | | | | | |
| | 2 | | | 100% | | | | | | | | | | | |
| | 3 | | | | | | | | | | | | | | |
| | 5 | | | 65% | | | | | | | | | | | |
| Patient 13 | 1 | | | 100% | | | | | | | | | | | |
| | 2 | | | 70% | | | | | | | | | | | |
| | 3 | | | 70% | | | | | | | | | | | |
| | 4 | | | 65% | | | | | | | | | | | |
| | 5 | | | 100% | | | | | | | | | | | |
| | 6 | | | 75% | | | | | | | | | | | |

**Figure 3.** Sequencing data from CTCs. Displayed values show the variant allele frequency. Empty boxes indicate that no mutation was detected.

## 4. Discussion

Mutations detected in cancer can help to choose an appropriate targeted therapy. In the advanced disease, analysis of a liquid biopsy can facilitate to identify such druggable targets. Various studies reported suitable tools for the analysis of single CTCs' WGA-products, either by investigation of single genomic positions by e.g., Sanger sequencing [22,23] or by sequencing multiple genomic positions by, e.g., NGS panels [24,25]. However, most available assays are either cost intensive, not adapted to single cell whole genome amplified DNA, or cover only single or not all relevant genomic positions for therapy selection in breast cancer. Here, we developed and provided the protocol for a multiplex PCR-based, Ampli1 WGA-adapted NGS panel that covers mutation hotspots in the *PIK3CA*, the *ESR1*, the *AKT1*, and the *ERBB2* genes that are most relevant for a liquid biopsy-based targeted therapy in breast cancer.

The assay was validated with spiked cell line cells and finally applied to CTCs from metastatic breast cancer patients. Mutations that provide the basis for potential targeted therapy were detected in 10 out of 13 patients. Although one CTC from patient 4 was found to harbor the *ERBB2* A775T mutation, it was not considered as clinically targetable since this mutation has not been characterized in breast cancer, yet. For the treatment of hormone receptor-positive, HER2-negative advanced breast cancers harboring *PIK3CA* variants such as N345K, E545K, H1047L, or H1047R, alpelisib plus fulvestrant is indicated [26]. Furthermore, HER2-positive tumors of patients with this variant exhibited resistance to trastuzumab [27]. The detected *ESR1* L536P mutation leads to a conformational change of the protein and to a ligand-independent ERα activation, thereby conferring resistance to aromatase inhibition [13]. However, treatment with second-generation selective ERα degraders such as fulvestrant is recommended [28]. The *AKT1* E17K mutation hyperactivates the mTOR pathway that could be targeted by mTOR inhibitors, such as everolimus, or AKT-inhibitors, such as ipatasertib and capivasertib [29,30]. Mutations such as S310F, L755S, or V777L in the *ERBB2* gene offer to be treated with the tyrosine kinase inhibitor neratinib [16].

The frequency of *PIK3CA* and *AKT1* mutations in our patient cohort exceeded the frequencies published in literature. *AKT1* E17K mutation frequencies between 1.4% and 8.2% have been described in breast cancer tissue [31] and mutations of the *PIK3CA* gene are found in 25%–40% of all breast cancers [32,33]. One reason for that is that such mutations can be acquired during the metastasis process leading to a discrepancy of the mutation status of the primary tumor and a liquid biopsy [34]. Furthermore, mutations in the *PIK3CA* and the *AKT1* gene lead to an increased proliferation of the tumor combined with reduced apoptosis [10,15] and might thereby cause a higher probability of CTCs detected in general and of CTCs with a high genome integrity in particular.

Most of the mutations we detected were located in well-described hotspot regions. Some mutations, however, are novel and have not been reported yet, such as the *PIK3CA* M318T, *ERBB2* L313P, or the *ERBB2* A775T mutations. These mutations were only detected in single WGA products and due to prior WGA, a false positive result cannot be totally excluded. To further investigate these mutations, tools to predict the effect on the protein structure have been applied [35–37]. Although they are not fully consistent, the results indicate that these mutations might affect the function of the protein (Table S3).

Heterogeneity was observed within CTCs from almost all patients. Such heterogeneity indicates that different subclones exist within the tumor and that the patient might therefore benefit from the combination of treatments targeting different proteins. However, it cannot be excluded that the observed assumed heterogeneity may at least in part be caused by false negativity due to allelic dropout during WGA. Thus, the absence of mutations does not necessarily mean that these mutations were not present in the initial cells. One example for allelic dropout is the T47-D reference cell 3: in this cell, only the mutated allele was amplified during WGA. This effect of allelic dropout is limiting all approaches for the analysis of DNA from single cells. The probability of allelic dropout increases with reduced DNA integrity. To minimize it, we focused on the analysis of WGA products that showed at least three bands in the quality control PCR.

Most studies on the clinical utility on mutations have been performed on ctDNA. However, the clinical utility of ctDNA and CTCs has not been compared so far and the analysis of CTCs and ctDNA might be complementary [34].

Altogether, we have developed a time and cost effective multiplex PCR-based assay for the analysis on a MiSeq system. Our assay covers the most relevant positions indicative for targeted therapies in breast cancer and can thereby improve therapy decision and help to improve the treatment of cancer patients eventually.

**Supplementary Materials:** The following are available online at http://www.mdpi.com/2076-3417/10/10/3364/s1, Table S1: Primer concentrations used in first PCR, Table S2: PCR protocols, Figure S1: Validation of first PCR, Figure S2: Distribution of reads to the analyzed fragments, Table S3: Prediction of effect of detected amino acid substitutions.

**Author Contributions:** A.F., T.F., and H.N. conceived the project and provided project leadership. A.F. and H.N. wrote the manuscript. M.R. and L.Y. contributed to single cell isolation. B.J. and N.K. contributed to sample collection. E.H. and D.N. contributed to data analysis. All authors read and approved the final manuscript.

**Funding:** AF was supported by the Duesseldorf School of Oncology (funded by the Comprehensive Cancer Centre Duesseldorf/Deutsche Krebshilfe and the Medical Faculty of the Heinrich Heine University Duesseldorf).

**Conflicts of Interest:** The authors declare no conflict of interest.

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
