# Peer review of "A Multiplex PCR-Based Next Generation Sequencing-Panel to Identify Mutations for Targeted Therapy in Breast Cancer Circulating Tumor Cells"

_applsci, doi:10.3390/app10103364_

Round 1

Reviewer 1 Report

This is a clearly written and concise manuscript in the hot area of liquid biopsies, which are emerging as an important tool in Oncology; thus, the development of novel time- and cost- effective platforms is of great interest.

The authors have developed such a tool and have provided adequate evidence on its analytical and, potentially, clinical validity.

Minor comments

  1. The authors are calling their assay "cost effective", but this is not explained in the manuscript. I believe it would be of interest to justify this claim by providing some numbers and comparing the cost/sample of their assay with others.
  2. Even though there are several published papers investigating mutations in CTCs in breast cancer, the authors do not discuss them. I think this is an important part missing from the Discussion section and it should be included along with some comparison between the different assays and how theirs is better (e.g. it is more time- or cost- effective).
  3. Analysis of corresponding tumor specimens would strengthen the clinical validity of the results. But, this is not a point I insist on.

Author Response

Dear Reviewer,

Thank you very much for reviewing our manuscript so carefully and for providing such valuable input and suggestions to improve its quality.

Below please find our replies and comments to the specific questions:

  1. The authors are calling their assay "cost effective", but this is not explained in the manuscript. I believe it would be of interest to justify this claim by providing some numbers and comparing the cost/sample of their assay with others.

The assay we present here enables detection of the most relevant mutations for targeted therapy in breast cancer for approximately 5 $ per whole genome amplification product of a single CTC. Costs of the assay are transparent to the reader: All required materials and sequencing capacity were listed in the materials and methods section. To the best of our knowledge, no commercially available assay for a similar approach and even approximately similar costs exists. For non-commercial assays, however, required materials and sequencing capacities are often not presented transparently which challenges a comparison of cost/sample.

  1. Even though there are several published papers investigating mutations in CTCs in breast cancer, the authors do not discuss them. I think this is an important part missing from the Discussion section and it should be included along with some comparison between the different assays and how theirs is better (e.g. it is more time- or cost- effective).

A section briefly addressing limitations of available assays was added to the discussion. Most commercially available assays for mutation analysis are either not adapted to single cell whole genome amplified DNA with Ampli1 approach, or cover only single or not all relevant genomic positions for therapy selection in breast cancer. The Ampli1 OncoSeek panel for example – advertised as the “first and only panel designed for single cell analysis” (Menarini Silicon Biosystems) – does not cover the ESR1 gene which is highly relevant for therapy decision in breast cancer. To compare our assay with other non-commercial assays – most of them only focus on mutations in one gene or do not target similar mutations as ours – a study comparing the assays on identical biological material needs to be performed which was not the aim of this manuscript and should be the topic of larger consortia.

  1. Analysis of corresponding tumor specimens would strengthen the clinical validity of the results. But, this is not a point I insist on.

We agree that the analysis of primary tumor tissue could be interesting to confirm that some of the detected mutations have been acquired after removal of the primary tumor. However, this was not the aim addressed here and will most likely not provide novel data. Moreover, tissue biopsies would be available from only few patients.

Yours sincerely,

Prof. Dr. Hans Neubauer

Reviewer 2 Report

Dear Editor in chief:

Applied sciences Journal

Regarding Manuscript: applsci- 784355

`` A multiplex PCR-Based Next Generation Sequencing-Panel to Identify Mutations for Targeted Therapy in Breast Cancer Circulating Tumor Cells`` by André Franken and coworkers:

Authors have separated circulating tumor cells (CTCs) from peripheral blood of breast cancer patients and by a multiplex PCR-based next generation sequencing-panel, they have amplified the whole genome and sequenced genes of interest. Using Cell Search analyses, they find the presence of 6 to 15340 CTCs per 7.5 ml of patients’ blood.  Their data shows the existence of different mutations in different CTCs of one patients as well as all 13 patients. Some of these mutations have been shown before and some were new. The manuscript (MS) has been written well and data are interesting.

Major comments:

  • Some mutations in Table (Figure) 3 are only present in one patient. Is there any possibility of mistake in sequencing? Describe.
  • To confirm the data, why authors did not sequence the main tumor in each patient to confirm the data or compare the data?
  • If I understood correctly, there is no similar mutation between reference cell lines and CTCs of patients. Why?

Minor comments:

  • There are 2 Figures 2 in the MS. Probably the last one should be Figure 3. These 2 Figures are similar to Table, not Figure. It is better to be changed to Tables, and depict them again for better quality.

Author Response

Dear Reviewer,

Thank you very much for reviewing our manuscript so carefully and for providing such valuable input and suggestions to improve its quality.

Below please find our replies and comments to the specific requests:

>Some mutations in Table (Figure) 3 are only present in one patient. Is there any possibility of mistake in sequencing? Describe.<

We agree that artefacts cannot be excluded, especially because of prior whole genome amplification although the error rate of the applied polymerase is as low as 0.33×10-6 per nucleotide. This is especially relevant for the novel mutations and has already been discussed in the manuscript. However, mutations relevant for targeted therapy selection were detected in more than one CTC in most cases. Thus, sequencing mistakes are highly unlikely in such cases. Moreover, the validity of the assay has been shown by the analysis of cell line cells displaying exclusively the expected mutations.

>To confirm the data, why authors did not sequence the main tumor in each patient to confirm the data or compare the data?<

The main goal of this manuscript is to present a tool to identify mutations highly relevant for targeted therapy in breast cancer. It is well known that such mutations can often not be detected in the primary tumor, which is why the analysis of liquid biopsies is an appealing concept to select drugs in the advanced disease. Thus, analysis of the primary tumor tissue is not suitable to confirm the data presented here.

>If I understood correctly, there is no similar mutation between reference cell lines and CTCs of patients. Why?<

This statement is not correct. The PIK3CA mutations E545K (MCF-7) and H1047R (T-47D) have been detected in CTCs from patients 12 and 13 and patients 3, 5, and 8, respectively. The AKT1 mutation E17K (long term cultured CTCs) has been found in CTCs from patients 2, 3, and 6. The ESR1 mutation E380Q (long term cultured CTCs) was, however, not detected in CTCs from our patient cohort although it is one of the most frequently observed and most relevant mutations for endocrine resistance in the ESR1 gene.

>There are 2 Figures 2 in the MS. Probably the last one should be Figure 3. These 2 Figures are similar to Table, not Figure. It is better to be changed to Tables, and depict them again for better quality.<

Figure 3 legend has been fixed. We agree to the suggestion that figure 2 and 3 could be shown as tables and will discuss with the editor whether colored tables are allowed.

Yours sincerely,

Prof. Dr. Hans Neubauer

Reviewer 3 Report

The manuscript submitted by Franken and colleagues encloses a very interesting work aimed at developing a multiplex PCR-based next generation sequencing-panel to identify mutations on single CTCs from breast cancer patients. Written in an elegant and clear style, this report could be readily applied into clinical practice with a high rate of success. 

Minor points to be addressed:

1. The authors must highlight in the Discussion section the advantages/disadvantages of their platform compared with other existing/reported platforms with the same (or similar) goal. Examples: DOI: 10.1039/c9lc01248f, DOI: 10.1111/cas.14092, as well as others that authors must identify.

2. Please, fix Figure 3 legend, which was shown as Figure 2. 

Author Response

Dear Reviewer,

Thank you very much for reviewing our manuscript so carefully and for providing such valuable input and suggestions to improve its quality.

Below please find our replies and comments to the specific requests:

  1. The authors must highlight in the Discussion section the advantages/disadvantages of their platform compared with other existing/reported platforms with the same (or similar) goal. Examples: DOI: 10.1039/c9lc01248f, DOI: 10.1111/cas.14092, as well as others that authors must identify.

A section briefly addressing limitations of available assays was added to the discussion. Most assays for mutation analysis are either cost intensive (e.g. 10.1111/cas.14092), not adapted to single cell whole genome amplified DNA, or cover only single (e.g. 10.1039/c9lc01248f) or not all relevant genomic positions for therapy selection in breast cancer. To the best of our knowledge no comparable commercially available assay for the analysis of the most relevant genomic positions for targeted therapy in breast cancer for approximately 5 $ per whole genome amplification product of a single CTC exists. The Ampli1 OncoSeek panel for example – advertised as the “first and only panel designed for single cell analysis” (Menarini Silicon Biosystems) – does not cover the ESR1 gene while costing a multiple. To compare our assay with other non-commercial assays – most of them only focus on mutations in one gene or do not target similar mutations as ours – a study comparing the assays on identical biological material needs to be performed which was not the aim of this manuscript and should be the topic of larger consortia.

  1. Please, fix Figure 3 legend, which was shown as Figure 2.

Figure 3 legend has been fixed.

Yours sincerely,

Prof. Dr. Hans Neubauer